# High Performance Thin Layer Chromatography (HPTLC) Analysis of Anti-Asthmatic Combination Therapy in Pharmaceutical Formulation: Assessment of the Method’s Greenness and Blueness

**DOI:** 10.3390/ph17081002

**Published:** 2024-07-29

**Authors:** Huda Salem AlSalem, Faisal K. Algethami, Maimana A. Magdy, Nourudin W. Ali, Hala E. Zaazaa, Mohamed Abdelkawy, Maha M. Abdelrahman, Mohammed Gamal

**Affiliations:** 1Department of Chemistry, College of Science, Princess Nourah bint Abdulrahman University, P.O. Box 84428, Riyadh 11671, Saudi Arabia; husalsalem@pnu.edu.sa; 2Department of Chemistry, College of Science, Imam Mohammad Ibn Saud Islamic University (IMSIU), P.O. Box 90950, Riyadh 11623, Saudi Arabia; falgethami@imamu.edu.sa; 3Pharmaceutical Analytical Chemistry Department, Faculty of Pharmacy, Beni-Suef University, Alshaheed Shehata Ahmad Hegazy St., Beni-Suef 62514, Egypt; 4Pharmaceutical Analytical Chemistry Department, Faculty of Pharmacy, Cairo University, Kasr-El-Aini, Cairo 11562, Egypt; hala.zaazaa@pharma.cu.edu.eg (H.E.Z.); dr.mohabdelkawy@hotmail.com (M.A.)

**Keywords:** Hydroxyzine Hydrochloride, Ephedrine Hydrochloride, Theophylline, TLC-densitometry, GAPI, AGREE, BAGI

## Abstract

A cost-effective, selective, sensitive, and operational TLC-densitometric approach has been adapted for the concurrent assay of Hydroxyzine Hydrochloride (HYX), Ephedrine Hydrochloride (EPH), and Theophylline (THP) in their pure powder and pharmaceutical forms. In the innovative TLC-densitometric approach, HYX, EPH, and THP were efficaciously separated and quantified on a 60F_254_ silica gel stationary phase with chloroform–ammonium acetate buffer (9.5:0.5, *v*/*v*) adjusted to pH 6.5 using ammonia solution as a mobile liquid system and UV detection at 220 nm. The novel TLC method validation has been performed in line with the international conference for harmonization (ICH) standards and has been effectively used for the estimation of the researched medicines in their pharmaceutical formulations without intervention from excipients. Additionally, parameters affecting the chromatographic analysis have been investigated. The new TLC approach’s functionality and greenness were appraised using three modern and automated tools, namely the Blue Applicability Grade Index (BAGI), the Analytical Greenness metric (AGREE), and the Green Analytical Procedure Index (GAPI) tools. In short, the greenness characteristics were not achieved as a result of using mandatory, non-ecofriendly solvents such as ammonia and chloroform. On the contrary, the applicability and usefulness of the novel TLC approach were attained via concurrent estimation for the three drugs using simple and straightforward procedures. Moreover, the novel TLC method outperforms previously published HPLC ones in terms of the short run time per sample and moderate pH value for the liquid system. According to the conclusions of comparisons with previously recorded TLC methods, our novel HPTLC method has the highest AGREE score, so it is the greenest HPTLC strategy. Moreover, its functionality and applicability are very appropriate because of the simultaneous assessment of three drugs in one TLC run. Furthermore, no tedious and complicated extraction and evaporation processes are prerequisites.

## 1. Introduction

The chemical name of hydroxyzine hydrochloride (HYX) is (RS)-2-[2-[4-[(4-chlorophenyl)phenylmethyl]piperazin-1-yl]ethoxy] ethanol dihydrochloride [1]. HYX is a diphenylmethane and piperazine class first-generation antihistamine (Figure 1). It is stated to have strong anxiolytic and mild anti-obsessive as well as antipsychotic properties [2]. The chemical name of Ephedrine hydrochloride (EPH) is (1R,2S)-2-(methylamino)-1-phenylpropan-1-ol hydrochloride [1], as seen in Figure 1. It is classified as a sympathomimetic amine and is commonly used for various medical purposes. EPH is used to treat hypotension related to anesthesia, as well as for decongestant and appetite suppressant purposes [3]. The chemical name of Theophylline (THP) is 1,3-dimethyl-3,7-dihydro-1H-purine-2,6-dione [1], as seen in Figure 1. It is used for respiratory diseases such as relaxing bronchial smooth muscle and asthma [4].

Both the British Pharmacopeia (BP) [1] and the United States Pharmacopeia (USP) [5] provide titration assays for quantifying HYX, EPH, and THP, independently, in raw materials [6]. HYX, EPH, and THP have been determined either alone or in combination with other components by HPLC [7,8,9,10,11,12,13,14,15,16,17,18,19,20], TLC-densitometry [20,21,22,23,24,25,26], GC [17,27,28], electrochemical [29,30,31,32,33,34], and different spectrophotometric methods [17,35,36,37,38,39]. After conducting a literature review, only two HPLC techniques were found for the simultaneous measurement of HYX, EPH, and THP [40,41].

Pharmaceutical formulation testing often involves the use of multianalyte analysis to determine the concentration and presence of different excipients and active pharmaceutical ingredients (APIs). This analysis is an essential part of quality control protocols, as the validity and effectiveness of pharmaceutical products depend on it [42].

Pharmaceutical research and industry often use thin-layer chromatography (TLC) as a powerful technique for analyzing and resolving chemicals in pharmaceutical formulations. TLC provides several advantages for drug resolution, including ease of use, affordability, quick analysis, adaptability, effective separation, sample recovery, minimal sample size requirements, and applicability for developing and optimizing analytical techniques [43].

No reliable TLC approach has been designed for the assay of the three medications under study simultaneously, based on the prior literature scanning. This research work aims to provide a cost-effective, sensitive, selective, and reliable TLC-densitometric approach for the concurrent assay of HYX, EPH, and THP using the same chromatographic environment. Moreover, TLC-densitometry can be utilized in quality control units as an inexpensive and time-efficient alternative to the costly HPLC method [44]. Compared to published HPLC methods [40,41], the new TLC-densitometric method does not necessitate expensive solvents or specialized equipment to separate the three drugs without overlapping with each other or with tablet additives. An additional aim of this project was to present a mini review of all reported TLC methods for the investigated drugs, with in-depth details about stationary phase materials, mobile phase systems, detection approaches, and greenness aspects using the AGREE tool in a comparative table.

## 2. Results and Discussion

For qualitative and quantitative drug assays and daily quality control, planar TLC chromatographic techniques with correct volume applications and computerized and automated determinations of the generated chromatograms have been shown to be dependable tools [45]. Drug mixes can be determined through the use of TLC-densitometry, which is an effective method for resolution. This method provides an easy approach to measuring the optical intensity of the isolated bands straight on TLC plates [46,47].

The principal mission of this research paper is to present a selective, reliable, and sensitive TLC-densitometric approach for the assessment of HYX, EPH, and THP in their raw materials and in their pharmaceutical formulation using the same chromatographic settings with convenient precision and accuracy.

### 2.1. Method Development and Optimization

Different factors involved during method development were investigated and optimized to enhance method performance in terms of separation, reliability, and functionality. The studied factors are listed as follows.

#### 2.1.1. Mobile Phase

Firstly, many green solvents were tested as developing systems, e.g., water and ethanol. However, inconvenient resolution and symmetry were obtained.

Various developing systems with different compositions have been investigated, such as hexane–ethyl acetate (6:4, *v*/*v*), hexane–methyl alcohol (8:2, *v*/*v*), and chloroform–ethyl acetate (8:2, *v*/*v*). Unfortunately, the three medicines were not moved from the baseline mark. In contrast, when the polarity of the liquid system was increased using chloroform–methyl alcohol (6:4, *v*/*v*), each of the three medications was moved to the front line. Consequently, it was required to use a liquid system with intermediate polarity, such as chloroform–methyl alcohol (9:1, *v*/*v*). However, a noteworthy decrease in the R_f_ value of EPH was detected, and a very bad resolution was obtained between the HYX and THP peaks. On the other hand, the addition of 0.1 mL and 0.2 mL of each glacial acetic acid and ammonia solution (33%), respectively, to the last-mentioned mobile phase greatly improved the peak shape and resolution, while precipitation of the developing mobile phase was observed. So, the optimal developing system consisted of chloroform and ammonium acetate buffer (9.5:0.5, *v*/*v*) adjusted to pH 6.5 using an ammonia solution (33%). An acceptable chromatogram was achieved in terms of resolution and peak shape. The recorded retardation factors R_f_ for EPH, THP, and HYX were 0.15, 0.40, and 0.65, correspondingly, as illustrated in Figure 2.

The physicochemical characteristics of the studied medicines were considered during the method’s development to attain an adequate chromatogram with minimal trials. The recognized pKa values are HYX = 7.45, EPH = 9.6, and THP = 8.8 [41]. The selected pH value for chromatographic resolution (pH = 6.5) makes EPH (which has a higher pka) in the ionized form be retained on the polar stationary phase and less dissolved in the non-polar mobile phase, giving a low R_f_ value of 0.15. Furthermore, at the working pH (6.5), THP is slightly ionized (with a moderate pKa value) and was eluted secondarily with an R_f_ value of 0.4. Regarding HYX with a lower pKa value, it was in a non-ionized form and dissolved in the non-polar mobile phase (less retained on silica), thus it was eluted at a higher R_f_ value of 0.65. An ammonia solution should be added for the mobile phase, particularly to elute HYX as a free base form [41].

#### 2.1.2. Scanning Wavelength

Several scanning wavelengths, including 215 nm, 220 nm, and 254 nm, were tested. Notably, scanning at 220 nm produced the ideal results for all medicines, with chromatographic peaks that were more symmetrical and uniform and with the least amount of noise. Moreover, 220 nm produced the highest sensitivity for each of the three medicines.

#### 2.1.3. Slit Size

The optimal slit width for a TLC scan depends on the size of the separated bands, the sensitivity of the detection system, and the necessary resolution. To ensure that only the desired analyte signals are detected, the slit width must be smaller than the band size. If the slit is too wide, it may absorb interference signals from nearby bands, resulting in lower resolution and an inaccurate peak area. After different trials using variable slit sizes (5 × 0.45, 6 × 0.45, 5 × 0.3, and 6 × 0.3 mm), the chosen slit size was 6 × 0.3 mm, providing adequate scanning of the peak area without interfering with nearby peaks.

#### 2.1.4. Saturation Period

Before placing the TLC plate in a chromatographic tank for TLC, it is crucial to ensure that the developing system vapor is dispersed uniformly across the chamber. This is achieved by allowing enough time for the tank to saturate to achieve reliable and consistent separations. After experimenting with various saturation times, it was determined that a period of 20 min is ideal. However, the sensitivity and resolution of the method were not affected by the higher saturation period. Conversely, less compact bands with inadequate resolution have been generated by saturation times shorter than 20 min.

### 2.2. Validation of the TLC-Densitometric Method

The method was validated in line with the ICH Q2 (R1) standards for the validation of analytical procedures [48].

#### 2.2.1. Linearity and Range

The noted linear ranges were 0.4–1.8, 2.0–16, and 0.4–1.8 µg/band for HYX, EPH, and THP, respectively. HYX, EPH, and THP concentrations were computed via an application using the following formulas:P.A.HYX = 3.62 C_HYX_ + 1.17         R^2^ = 0.9997
P.A.EPH = 0.266 C_EPH_ + 0.463         R^2^ = 0.9997
P.A.THP = 3.94 C_THP_ + 0.269         R^2^ = 0.9998
where P.A. is the estimated peak area (×10^−3^). C_HYX_, C_EPH,_ and C_THP_ are the concentrations of HYX, EPH, and THP in µg/band, correspondingly. The R^2^ symbol refers to the coefficients of determination.

The near-unity values for the coefficients of determination (R²) normally indicate a good fit, and the models are accurately representing the data. Items for regression formulas and linear ranges are listed in Table 1.

#### 2.2.2. Quantitation and Detection Limit (LOQ and LOD) Calculations

The low stated values of both LOQ and LOD indicate great sensitivity for the TLC approach, as verified in Table 1.

#### 2.2.3. Accuracy

Detailed results of accuracy for the assay of pure powder samples of HYX, EPH, and THP using the novel HPTLC-densitometric approach are demonstrated in Appendix A. Furthermore, the convenient recoveries of the standard addition approach assured the accuracy of the HPTLC approach, as described in Appendix A.

Moreover, a statistical comparison of the outcomes for the novel TLC method and the reported HPLC method (RP-HPLC method using a C18 stationary phase and a mobile phase consisting of equal volumes of acetonitrile and 0.1% (*w*/*v*) aqueous ammonium carbonate buffer with pH 7 controlled with acetic acid at a flow rate of 2 mL/min) [40] demonstrates that the computed F- and t-values are less than the hypothetical ones, indicating that there are no noticeable distinctions between the two chromatographic techniques in terms of accuracy, as illustrated in Table 2.

#### 2.2.4. Precision

Repeatability values (% RSDs) were 1.1 for HYX, 1.3 for EPH, and 1.2 for THP (Table 1). Nine replicates (n = 9) of freshly prepared solutions of HYX, EPH, and THP, equivalent to 0.6, 0.8, and 1 μg/band for HYX and THP, and equivalent to 4, 6, and 8 μg/band for EPH, on three sequential days (inter-days) were analyzed using the novel TLC method, and the % RSDs of the recorded peak areas were 1.3 for HYX, 1.3 for EPH, and 1.5 for THP. The method’s precision is confirmed as all recorded %RSDs values are less than 2, as recommended by ICH validation protocols [48].

#### 2.2.5. Robustness

The results in Table 3 demonstrate that the proposed approach is adaptable to minor variations in mobile phase parameters: chloroform (9.5 mL ± 1%), ammonium acetate buffer (0.5 mL ± 0.5%), and pH (6.5 ± 0.1).

System integration evaluations are an essential component of procedures involving liquid chromatography [49,50]. They are used to verify that the reproducibility and resolution of the chromatographic arrangement are adequate for the planned assay. Considerations including peak symmetry, resolution (Rs), and selectivity factors (α) were assessed. Table 4 demonstrates the acceptable values for resolution, selectivity, and the symmetry factor. The resolution is consistently higher than two, and the selectivity is higher than one.

### 2.3. Application to Bronchaline^®^ Tablets: Dosage Form

The novel HPTLC method’s accuracy was tested for the analysis of HYX, EPH, and THP in Bronchaline^®^ tablets, as illustrated in Appendix A. The novel HPTLC method was efficaciously applied for the analysis of HYX, EPH, and THP in their pharmaceutical tablets, and by applying the standard addition protocol, no interference from additives that may be found in the tablets was observed, as exposed in Appendix A. The results of recoveries for the three drugs were around 100% with standard deviations less than 2, which confirmed the method’s accuracy in light of the ICH instructions.

### 2.4. The Greenness Assessment for the Novel TLC Method

To minimize the negative effects of chemical solvents on the environment and to enhance the general health of the planet, it is vital in the modern world to employ environmentally friendly chemicals and analytical methods. Concerning this TLC method, the authors tried to align with the values of green chemistry by using green solvents. However, a non-convenient chromatogram was obtained in terms of peak symmetry and overlap. Therefore, using chloroform was mandatory for the success of the novel TLC chromatographic method. Precisely, two computerized greenness evaluation tools were employed, i.e., the Analytical Greenness (AGREE) [52] and the Green Analytical Procedure Index (GAPI) [53] approaches.

In various articles [54,55,56,57,58,59,60,61,62], the dependability of the AGREE and GAPI tools for the appraisal of eco-friendly aspects is well verified. As seen in Figure 3, the TLC approach’s AGREE [52] score of 0.56 reflects the low greenness attributes. Notably, the three red sub-sections 3, 10, and 11 refer to the offline TLC situation, non-bio-based resources, and using hazard-mandatory chloroform.

Concerning GAPI outcomes, the resultant pictogram, Figure 4, contains four red subsections that are sections 1, 7, 13, and 15. These red sections refer to offline assay, the use of chloroform hazard solvent, the possibility of chloroform vapors being released into the environment, and the non-treatment of solvent waste.

### 2.5. Evaluation of the Method’s Practicality via the Blue Applicability Grade Index (BAGI) Approach

The new automated ten-parameter (BAGI) tool [63] was implemented to evaluate the methods’ usefulness and practicability. The BAGI tool provides an index score and an asteroid-like image. Herein, the overall BAGI score was 77.5, demonstrating the applicability and efficiency of the investigated HPTLC method, as illustrated in Figure 5. This favorable score results from many intense blue subsections that can be attributed to the simultaneous analysis of the three medicines in a single run, using an automated instrument. On the other hand, a single white subsection was observed (subcategory 4), resulting from the fact that only one sample could be prepared per chromatographic run.

In brief, the greenness characteristics were not achieved as a result of using non-ecofriendly solvents such as ammonia and chloroform. On the contrary, the applicability and usefulness of the novel TLC approach were attained via concurrent estimation for the three drugs using simple and straightforward procedures.

### 2.6. The Method’s Comparisons with Previously Reported HPLC and TLC Methods

Table 5 summarizes the comparison between the analytical parameters of the developed HPTLC method and those of previously reported HPLC methods in the literature. The oldest RP-HPLC method [40] for assaying mixtures of HYX, EPH, and THP was reported by Roberts and Delaney in 1982. Ten drops of concentrated NaOH solution were added to the tablet for the dissolution of the three compounds before dilution with the mobile phase. The liquid phase was composed of acetonitrile and 0.1% (*w*/*v*) ammonium carbonate buffer aqueous solution previously adjusted to pH 7 using acetic acid (50:50, *v*/*v*). Ammonia solution should be added to each sample to elute HYX. HYX was not eluted from the column in the absence of ammonia. It seems that HYX must be forced into its free base form with ammonia. The calculated resolution between the three peaks was greater than 3 in all cases. Acceptable recovery, accuracy, and precision results were documented. Meanwhile, the exact linearity ranges for the three studied medications were not stated. Likewise, the second reported HPLC method [41] assessed the three aforementioned drugs in addition to papaverine hydrochloride simultaneously. A gradient elution program and C18 column were used, and two detection wavelengths were assigned: 220 nm for EPH and HYX, while 240 nm was employed for THP and papaverine hydrochloride. The resolution between THP and EPH peaks was 1.475, which was acceptable according to the European Pharmacopoeia guidelines. The demerits of the reported HPLC method [41] included the very low pH value (2.4) of the liquid phase, which is not preferred in terms of the stability of the stationary phase and studied drugs. Additionally, it consumed large volumes of non-green acetonitrile as the overall run-time was relatively long (≃15 min) without including baseline normalization time. On the contrary, the novel HPTLC method provided a short run time per sample (≃2.5 min), including saturation time, and was conducted at a reasonable pH value (6.5). Moreover, the novel HPTLC approach assessed the three drugs at low quantitation limit levels.

Notably, the greenness characteristics of the novel HPTLC approach compared to the HPLC one highlight several implications for sustainable analytical practices in the pharmaceutical industry. The lower solvent consumption, energy efficiency, minimized waste generation and its disposal, cost-effectiveness, instrument accessibility, and versatility associated with HPTLC contribute to sustainable analytical practices. Implementing these practices can lead to reduced environmental impact, lower costs, and improved operational efficiency in the pharmaceutical industry [64,65].

In brief, the comparative Table 6 represents a mini review of all reported TLC methods for the investigated drugs, with in-depth details about stationary phase materials, mobile systems, detection approaches, and greenness aspects using the AGREE tool. No automated HPTLC methods were reported for the assay of hydroxyzine HCl. Therefore, it is not included in comparative Table 6. Regarding EPH analysis, the HPTLC-MS reported by Goyal and his colleagues [25] is considered the most specific method as it also determined analogues of ephedrine, such as pseudoephedrine, and phenylpropanolamine in forensic samples using an MS detector. However, its sensitivity [12.00–22.00 μg/band] is very low if compared to our novel approach [2.00–16.00 μg/band]. Regarding THP analysis, the HPTLC approach recorded by Devarajan and his team [24] is the most sensitive one, as it can detect concentrations as low as 20.00 ng/spot. However, this method detects THP in plasma alone and not in pharmaceutical dosage forms. Our novel HPTLC approach has the merit of controlling pH = 6.5 for the liquid mobile system, while other liquid systems did not control the pH value, which may affect the reproducibility of results. Notably, all reported TLC methods used normal aluminum-supported sheets of silica gel as a stationary material, while a variety of organic and aqueous solvents were utilized in the liquid system. All reported TLC methods in Table 6 use alcohols such as methanol, ethanol, propanol, and n-butanol as components of the liquid system, except our novel approach and the TLC method [23]. Concerning mobile systems for the TLC analysis of EPH, triethylamine or ammonia solution is employed in all reported TLC methods [20,25,26] and our method, while acetic acid was used in one TLC method [21] only for the assay of THP. The wavelength of 220 nm is reported for the first time for the determination of THP, which was previously detected at 274 ± 2 nm in all previous studies. However, EPH was previously analyzed at 220 nm in Deltarhino^®^ nasal spray [20]. Generally, the outcomes of validations and recoveries for the three drugs at 220 nm were adequate and supported the accuracy and reliability of the new approach for the concurrent assay of Hydroxyzine Hydrochloride (HYX), Ephedrine Hydrochloride (EPH), and Theophylline (THP) in their pure powder and pharmaceutical formulations.

According to the conclusions of Table 6, our novel HPTLC has the highest AGREE score, so it is the greenest HPTLC one. Moreover, its functionality and applicability are very appropriate, as indicated by its BAGI total score of 77.5 out of 100, because of the simultaneous assessment of three drugs in one TLC run. Furthermore, no tedious and complicated extraction and evaporation processes are prerequisites for their assays. Our new approach assessed the aforementioned three drugs in one chromatographic run in a real pharmaceutical formulation [Bronchaline^®^ tablets] in addition to the simplicity and availability of the instrument in most laboratories.

By applying the novel HPTLC method specifically for HYX, EPH, and THP, analysts can achieve better separation, detection, and quantification of these drugs. This would lead to more accurate results, improving the quality of analysis in pharmaceutical research. The new HPTLC method can enhance the quality control process by enabling more precise measurements of pure drugs and pharmaceutical formulations [66]. This would ensure that pharmaceutical products containing these substances meet the required specifications, reducing the risk of substandard or ineffective medications reaching asthmatic patients. Generally, this novel analytical approach would contribute to the pharmaceutical industry’s ability to produce effective, safe, and high-quality pharmaceutical formulations [67].

## 3. Materials and Methods

### 3.1. Analytical Instruments

A short UV radiation lamp that was designed to emit ultraviolet (UV) radiation at a specific wavelength of 254 nm (manufactured by Merckmillipore company in Burlington, MA, USA) was used for drug visibility in the primary trials. The quantitative studies were accomplished using a TLC scanner 3 densitometer (Camag Company, Muttenz, Switzerland). The spraying frequency was 10 µL/s. The slit size was controlled at 6 mm × 0.3 mm. The adjusted resolution of the data was 100 µm/step during all measurements. The scanning speed was fixed at 20 mm/s. A 100 µL syringe sample injector for TLC Linomat IV (Camag, Muttenz, Switzerland) was employed in all assessments. TLC plates covered with 60F254 silica gel in dimensions 20 × 20 cm (Fluka, Sigma-Aldrich Chemie Gmbh, Darmstadt, Germany) were used in all chromatographic runs.

### 3.2. Resources

(a)Pure standards

Hydroxyzine Hydrochloride, Ephedrine Hydrochloride, and Theophylline were generously provided by the Egyptian Chemical Industries Development (CID) Organization, Cairo, Egypt. Their purities were 100.52, 99.85, and 99.32% for HYX, EPH, and THP, respectively, as stated by the previously stated HPLC approach [40].

(b)Pharmaceutical preparation

Bronchaline^®^ tablets, produced by CID Company, Cairo, Egypt (Batch No. 14311W), are stated to have 10 mg of HYX, 15 mg of EPH, and 120 mg of THP.

(c)Solvents and Chemicals

All through this investigation, analytical-grade reagents and chemical substances were employed, lacking additional refining.

In this study, analytical-grade liquids and chemicals of purity not less than 99.00%, such as methyl alcohol, chloroform, ammonia solution (33%), and ammonium acetate powder, were used without additional refining. All the aforementioned chemicals and reagents were purchased from El-Nasr Pharmaceutical Chemical Organization, Abu-Zabaal, Al-Qalyubia, Egypt.

### 3.3. Preparation of Stock Solutions

Three different 100 mL capacity flasks were filled with 0.1 g of HYX, EPH, and THP (1 mg/mL in methyl alcohol); each flask was then filled with 50 mL of methyl alcohol, stirred to dissolve it, and the total amount was then completed with methyl alcohol.

### 3.4. Chromatographic Separation

Different aliquots of HYX, EPH, and THP were exactly measured from their stock solutions and relocated into a set of 10 mL volumetric flasks that were filled with methyl alcohol. Using a Camag Linomat IV injector, 10 µL of all solutions was spread in 6 mm wide × 0.3 mm bands on 20 × 10 cm plates with a 250 µm density. The drug’s bands were placed 10 mm from the sides and bottom and at intervals of 5 mm. A linearly rising chromatogram was created up to an 8 cm distance in a chromatographic container that had been saturated for 20 min with the developing fluid system, which is a mixture of chloroform and ammonium acetate buffer solution in a volume ratio of 9.5:0.5, which had been adjusted to pH 6.5 with 33% aqueous ammonia at 25 °C. At 220 nm, the integrated peak area (×10^−3^) was plotted against the relevant concentrations of the three medicines to create the calibration curves.

### 3.5. Method Validation

The method was validated in line with the ICH Q2 (R1) standards for the validation of analytical procedures [48].

#### 3.5.1. Linearity and Range

Volumes equivalent to 0.4–1.8, 2–16, and 0.4–1.8 mg from the aforementioned start solution (1 mg/mL) of HYX, EPH, and THP were transferred into two sets of 10 mL volumetric flasks, then completed to the final volume with methyl alcohol. Volumes of 10 μL of each prepared solution were applied to the TLC plates and developed under the stated settings in Section 2.1. The peak areas for HYX, EPH, and THP were recorded at 220 nm, and the calibration curves were created as the peak area versus the corresponding concentrations.

#### 3.5.2. Quantitation and Detection Limit (LOQ and LOD) Calculations

A variety of specimens of the three medicines in the spectrum of detection range—0.1, 0.2, and 0.3 μg/band for HYX and THP, and 0.5, 1.0, and 1.5 μg/band for EPH—were used to study specific curves in six separate investigations. LOQ and LOD were computed according to the formulas LOQ = 10 (SD/slope) and LOD = 3.3 (SD/slope).

#### 3.5.3. Accuracy

Comparing the measured results with an expected value is an accepted way of demonstrating accuracy. Accuracy is assessed through reference material comparison, spike studies, and parallel procedure comparison. It is determined as the mean percent recovery. As displayed in Appendix A, pure powdered samples for HYX, EPH, and THP were examined to verify accuracy. Using their regression mathematical models, the overall outcomes of the percent recoveries of the three medications in pure form were determined. Additionally, as shown in Appendix A, the accuracy of the HPTLC approach was evaluated in Bronchaline^®^ tablets by applying the standard addition methodology to known amounts of HYX, EPH, and THP standards that have been added at various concentration levels. Moreover, the new HPTLC method’s results were compared with those of a second, well-established HPLC technique [40].

#### 3.5.4. Precision

Precision describes the level of reproducibility or repeatability of analytical data obtained under specific scenarios. Repeatability measures precision over a short period of time under the same conditions. Intermediate precision assesses the precision between different analysts, different instruments, or different days. Precision is assessed through the standard deviations, relative standard deviations, and confidence intervals. Nine replicates (n = 9) of freshly prepared solutions of HYX, EPH, and THP, equivalent to 0.6, 0.8, and 1 μg/band for HYX and THP, and equivalent to 4, 6, and 8 μg/band for EPH, were studied to assess system precision. Intra-day repeatability was established by performing triplicate assessments of the solutions on the same day.

#### 3.5.5. Robustness

Robustness, which measures a method’s ability to withstand minor but intentional changes in its parameters, is a sign of the method’s dependability. Robustness is a measure of the analytical procedure’s capacity to meet the desired performance requirements during practical applications. Calculation and comparison of relative standard deviations are very useful to assess the method’s robustness. To assess the TLC method’s robustness, compositions of the mobile phase were changed, where pH 6.5 ± 0.1 of the developing system is a critical value for resolution and the peaks’ symmetry of the ternary mixture.

### 3.6. Application to Bronchaline^®^ Tablets: Dosage Form

Selecting 10 tablets for testing is a balance between obtaining a representative sample and ensuring the efficient use of resources in the QC process. Moreover, in all chapters of the British Pharmacopeia (BP) [1] that discuss the analysis of tablet dosage forms, they select 10 tablets as a sample size. Ten Bronchaline^®^ tablets were well crushed and mixed. The mass of one tablet was 0.26 g, and the mass of 10 tablets was 2.6 g. These 2.6 g of the grinded tablets contained 100 mg of HYX, 150 mg of EPH, and 1200 mg of THP. A carefully weighed amount of about 0.65 g of tablet powder, corresponding to 25 mg of HYX, 37.5 mg of EPH, and 300 mg of THP, was dissolved in 75 mL of methyl alcohol and shaken for 20 min in a sonicator. The mixture was filtered, and the clear solution was diluted in a 100 mL volumetric flask with methyl alcohol. A part of this resultant solution was utilized as a start standard solution for HYX (250 μg/mL) and for EPH (375 μg/mL), and another part was diluted with methyl alcohol in a ratio of 1:1 to obtain a start solution for THP (1.5 mg/mL). The developed procedures were then followed for the analysis of HYX, EPH, and THP, and the concentrations for the three medicines were obtained by interpolation in their equivalent regression formulas.

The novel HPTLC method’s accuracy was tested for the analysis of HYX, EPH, and THP in Bronchaline^®^ tablets.

## 4. Conclusions

An innovative, sensitive, and selective HPTLC-densitometric method is presented for the first time for the simultaneous determination of HYX, EPH, and THP mixtures in medicinal products and pure raw materials. In labs without an HPLC instrument, the developed TLC-densitometric approach can be efficiently used as an alternative to the HPLC procedures mentioned above. Since multiple samples may be performed synchronously with a small portion of the developing system, the novel TLC-densitometric approach saves time and money for the analysis. A statistical comparison of the outcomes for the novel TLC method and the reported HPLC method demonstrates that the computed F- and t-values were less than the hypothetical ones, indicating that there are no noticeable distinctions between the two chromatographic techniques in terms of accuracy. The low values for (%RSD) that are less than 1.5 in all measurements for the three drugs indicate appropriate results for precision and robustness. The functionality and applicability of the novel TLC method have also been proven by a 77.5 score in the Blue Applicability Grade Index (BAGI) approach. At last, we were able to demonstrate that there is no need for a preparatory separation step when using the proposed TLC approach for repetitive testing of the medicines under study.

## Figures and Tables

**Figure 1 pharmaceuticals-17-01002-f001:**
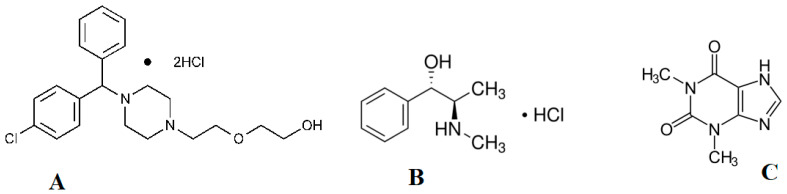
Chemical structure of (**A**) Hydroxyzine Hydrochloride (HYX), (**B**) Ephedrine Hydrocloride (EPH), and (**C**) Theophylline (THP).

**Figure 2 pharmaceuticals-17-01002-f002:**
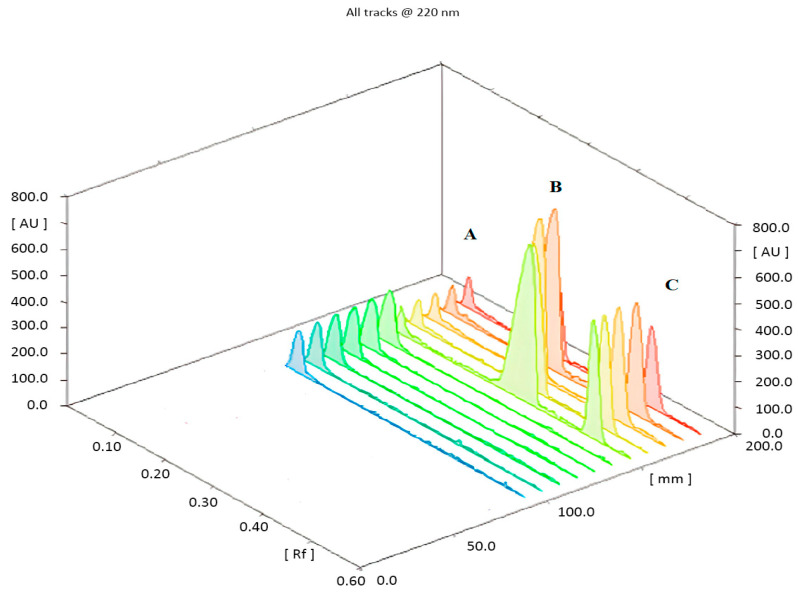
Thin-layer chromatogram of separated peaks of (A) Ephedrine Hydrocloride, (B) Theophylline, and (C) Hydroxyzine Hydrochloride, using chloroform–ammonium acetate solution (9.5:0.5, *v*/*v*) adjusted to pH 6.5 using 33% diluted ammonia solution as a developing system.

**Figure 3 pharmaceuticals-17-01002-f003:**
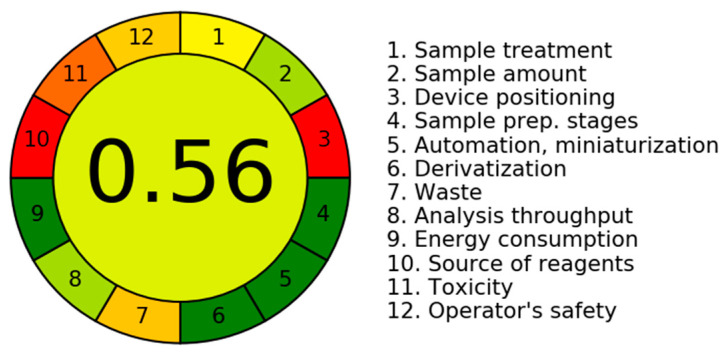
The AGREE pictogram for the assay of the triple mixture using the aforementioned TLC method with a mobile phase consisting of a mixture of chloroform and ammonium acetate solution (9.5:0.5, *v*/*v*) adjusted to pH 6.5 using diluted ammonia solution.

**Figure 4 pharmaceuticals-17-01002-f004:**
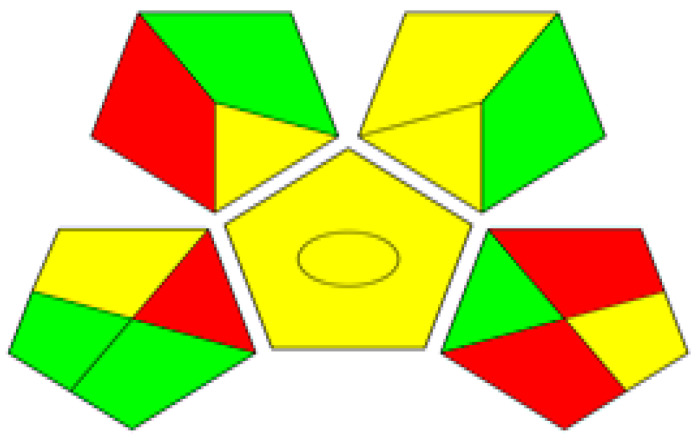
The GAPI pictogram for the assay of the triple mixture using the aforementioned TLC method with a mobile phase consisting of a mixture of chloroform and ammonium acetate solution (9.5:0.5, *v*/*v*) adjusted to pH 6.5 using diluted ammonia solution.

**Figure 5 pharmaceuticals-17-01002-f005:**
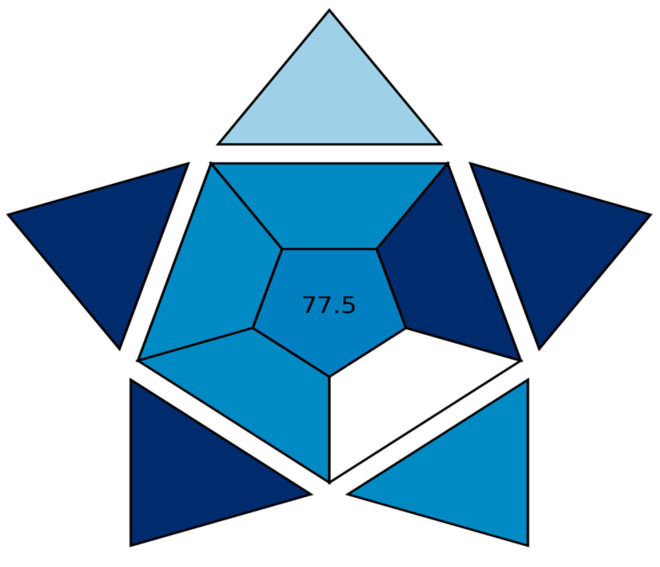
The BAGI pictogram and score for the assay of the triple mixture using the aforementioned TLC method with a mobile phase consisting of a mixture of chloroform and ammonium acetate solution (9.5:0.5, *v*/*v*) adjusted to pH 6.5 using diluted ammonia solution.

**Table 1 pharmaceuticals-17-01002-t001:** Outcomes of validation parameters for the novel TLC-densitometric method developed for the analysis of Hydroxyzine Hydrochloride, Ephedrine Hydrochloride, and Theophylline.

Parameter	TLC-Densitometric Method		
	HYX	EPH	THP
Range (μg/band)	0.40–1.80	2.00–16.00	0.40–1.80
Linearity			
Slope	3.62	0.266	3.94
Intercept	1.17	0.463	0.269
The coefficient of determination (R^2^)	0.9997	0.9997	0.9998
Accuracy (Mean ± SD)	99.4 ±1.1	99.7 ±1.3	99.8 ±1.2
Precision Repeatability * Intermediate precision * Calculated as (%RSD)	1.1 1.3	1.3 1.3	1.2 1.5
LOD (μg/band) ^a^	0.09 ^b^	0.52	0.11
LOQ (μg/band) ^a^	0.27 ^c^	1.59	0.31

* The average of three distinct concentrations analyzed in triplicate during the day is the intraday precision (*n* = 3). The interday precision (n = 3) is the three-day average of three distinct concentrations analyzed in triplicate. ^a^ Considered from the formulas [LOD = 3.3 (SD/slope), LOQ = 10 (SD/slope)]. For HYX as an example, LOD ^b^ = 3.3 × 0.0988/3.62 = 0.09, and LOQ ^c^ = 10 × 0.0988/3.62= 0.27.

**Table 2 pharmaceuticals-17-01002-t002:** Statistical comparisons between the new TLC-densiometric method and the reported HPLC one for the analysis of HYX, EPH, and THP in their pharmaceutical formulations.

Items	Bronchaline^®^ Tablets			Reported Method [40] *		
	HYX	EPH	THP	HYX	EPH	THP
Mean **	96.7	95.8	105.0	97.3	96.2	104.6
SD	1.4	1.2	1.2	1.2	1.5	1.2
%RSD	1.5	1.3	1.1	1.3	1.5	1.1
n	6	6	6	6	6	6
Variance	2.042	1.560	1.366	1.493	2.167	1.471
Student’s *t*-test (2.228) ***	2.162	1.427	0.363	—	—	—
F-value (5.050) ***	1.367	1.388	1.077	—		

* The RP-HPLC method using a C18 column and a mobile phase consisting of equal amounts of acetonitrile and 0.1% (*w*/*v*) aqueous ammonium carbonate buffer solution adjusted to pH 7 with acetic acid at a flow rate of 2 mL/min. ** Average of 3 measurements. *** The comparable tabulated values of t and F at *p* = 0.05 are indicated by the value between parentheses.

**Table 3 pharmaceuticals-17-01002-t003:** Experimental results of robustness for the determination of hydroxyzine hydrochloride, ephedrine hydrochloride, and theophylline by the novel HPTLC-densitometric approach.

Parameters (%RSD)	TLC-Densitometric Method		
	HYX	EPH	THP
Chloroform (9.5 mL ± 1%)	0.397	0.547	0.723
Ammonium acetate buffer (0.5 mL ± 0.5%)	1.124	1.217	0.945
pH of the developing system (6.5 ± 0.1)	0.531	0.458	0.421

**Table 4 pharmaceuticals-17-01002-t004:** System suitability testing parameters of the HPTLC-densitometric method for the determination of hydroxyzine hydrochloride, ephedrine hydrochloride, and theophylline.

Parameters	TLC-Densitometric Method			Reference Value [51]
	EPH	THP	HYX	
Selectivity factor (α)	2.67	1.62	1.62	>1
Resolution (R)	5.11		5.83	R > 1.5
Symmetry factor “Tailing factor” (T)	1.12	1.05	1.14	≈1

**Table 5 pharmaceuticals-17-01002-t005:** Comparison between the proposed HPTLC method and the reported methods for the determination of HYX, EPH, and THP.

Parameter	Proposed HPTLC Method	Reported RP-HPLC Method I [40]	Reported HPLC Method II [41]
Mobile phase	Chloroform–ammonium acetate buffer (9.5:0.5, *v*/*v*)	Ammonium carbonate buffer-acetonitrile (50:50 *v*/*v*)	Water containing phosphoric acid -acetonitrile (gradient elution)
Detection wavelength	220 nm	254 nm	220 nm for EPH and HYX and 240 nm for THP and papaverine hydrochloride
Linearity	HYX (0.40–1.80 μg/band) EPH (2.00–16.00 μg/band) THP (0.40–1.80 μg/band)	Not assigned	HYX (50–150 μg/mL) EPH (100–300 μg/mL) THP (500–1500 μg/mL)
Run time/per sample	≃2.5 min [Elution time + saturation time/number of samples per plate]	≃12 min Without including the actual time required for normalizing the baseline	≃15 min Without including the actual time required for normalizing the baseline
pH	6.5	7	2.4
Comments	Short run time per sample Moderate pH value Low quantitation limits	Ammonia should be added to each sample to elute HYX Linearity range for mixture analysis not applicable	Very low pH value Moderately long run time Papaverine hydrochloride was also assessed

**Table 6 pharmaceuticals-17-01002-t006:** Comparisons between the novel HPTLC method and the previously related ones for EPH and THP in the literature.

Reference Numbers	[20]	[25]	[26]	Our Novel Method	[21]	[22]	[23]	[24]	Our Novel Method
Drug Name	Ephedrine Hydrochloride (EPH)	Theophylline (THP)
Other analyzed drugs	naphazoline nitrate and methylparaben	analogues of ephedrine, such as pseudoephedrine, and phenylpropanolamine	carbinoxamine, pholcodine	Hydroxyzine HCl and Theophylline	caffeine as a probable interfering drug and paracetamol as internal standard	No other drugs were assessed	caffeine and theobromine	No other compounds were assessed	ephedrine HCl and Theophylline
Linear range (µg/band)	2.00–16.00	12.00–22.00	5.00–45.00	2.00–16.00	80.00–160.00 ng/spot	0.50–2.000 μg mL^−1^	250.00–1500.00 ng/spot	20.00–100.00 ng/spot	0.40–1.80
Retardation factor time (R_f_)	0.15	0.41	0.12	0.15	Not recorded	0.48	0.25	0.54	0.40
Mobile phase	ethyl acetate–ethyl alcohol–tri ethylamine (8.0:2.0:0.2, in volumes)	n-butyl acetate–acetone–n-butanol–5M NH_4_OH: methanol (4:2:2:1:1, in volumes)	Propanol–Chloroform–ammonia (4:6:0.1, in volumes)	Chloroform–ammonium acetate buffer (9.5: 0.5, *v*/*v*) adjusted to pH 6.5 using ammonia solution 33%	Toluene–isopropanol–acetic acid (8:2:0.5 by volumes)	Chloroform–methyl alcohol (9:1, *v*/*v*)	Acetone–toluene–chloroform (4:3:3, in volumes)	Chloroform–methyl alcohol 9:1 *v*/*v*	Chloroform–ammonium acetate buffer (9.5: 0.5, *v*/*v*) adjusted to pH 6.5 using ammonia solution 33%
Stationary material	silica gel aluminum normal sheets (20 × 10 cm and 250 μm thickness)	Aluminum-supported sheets of Silica gel 60 F₂₅₄	Aluminum-supported sheets of Silica gel 60 F₂₅₄	TLC plates covered with Silica gel 60 F₂₅₄	Silica gel 60 F₂₅₄ sheets	Silica gel 60 F₂₅₄ sheets	Fluorescence indicator-equipped LiChrospher silica gel sheets	Silica gel 60 F₂₅₄ sheets	Silica gel 60 F₂₅₄ plates
Detection method	UV at 220.00 nm	HPTLC–MS	UV at 245.00 nm	UV at 220.00 nm	UV at 278.00 nm	UV at 277.00 nm	UV at 274.00 nm	UV at 272.00 nm	UV at 220.00 nm
Greenness appraisal using AGREE approach	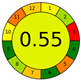	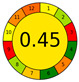	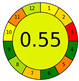	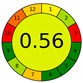	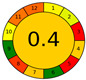	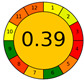	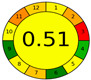	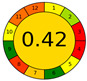	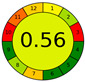
Merits	Assay of real pharmaceutical [ Deltarhino^®^ nasal spray] Final analytical eco-scale score of 71 referring to convenient greenness aspects Simplicity and availability of instrument in most laboratories	This technology determined ephedrine analogues with little sample preparation, a rapid and simple process, and no derivatization Applied for suspected forensic specimens	Applied for assay of Antitussive Cyrinol^®^ Syrup Well-resolved bands with excellent symmetry Simplicity and sensitivity of the method No interference of excipients	Assay of real pharmaceutical [Bronchaline^®^ tablets] Assay of three drugs in one run Relative greenness aspects Simplicity and availability of instrument in most laboratories Convenient method functionality according to BAGI score of 77.5	Applied efficiently for assay of THP in human plasma No interference was recorded from probable interfering drugs, e.g., caffeine and paracetamol High sensitivity in nanoscale	Assay of THP in post-mortem blood in 60 min It requires less solvent and does not necessitate a lengthy cleanup process The extreme sensitivity No interference was recorded from probable interfering drugs, e.g., ephedrine, caffeine, salbutamol, and paracetamol	Applied positively for assay THP in Mate soft drinks and Mate beer Very sensitive [ limit of quantitation was 4 ng/zone] Convenient recoveries, accuracy, and precision outcomes Practical and useful methods as it assayed 3 drugs in a single run	Measured the plasma levels of theophylline following the administration of a single, commercially available, oral sustained-release tablet The most sensitive HPTLC method for assay of THP in clinical and pharmacokinetic studies Rf values of Potential metabolites in plasma were reported	Assay of real pharmaceutical [Bronchaline^®^ tablets] Careful adjustment of pH Assay of three drugs in one run Relative greenness aspects Simplicity and availability of instrument in most laboratories Convenient method functionality according to BAGI score of 77.5
Demerits	No applications in serum or urine No assessment of impurities	HPTLC-MS instrument is expensive and not available in the majority of pharmaceutical labs Not applied for pharmaceutical formulation	No applications in serum or urine No assessment of impurities	No applications in serum or urine No assessment of impurities	25 min migration time [long time] Deproteinization and many extraction and evaporation steps Non-green method according to AGREE score	Less functional and practical method as one drug assayed per one run Not applied for impurity detection and pharmaceuticals Many tedious extraction and evaporation procedures	Not applied for impurity detection and pharmaceutical assay No applications in serum or urine Long degassing time 45 min	Less functional and practical method as one drug assayed per one run Very complicated extraction and evaporation procedures Not applied for impurity detection and pharmaceutical assay	No applications in serum or urine No assessment of impurities

## Data Availability

The corresponding scholar will provide the necessary information as needed.

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
