# Peer review of "High Performance Thin Layer Chromatography (HPTLC) Analysis of Anti-Asthmatic Combination Therapy in Pharmaceutical Formulation: Assessment of the Method’s Greenness and Blueness"

_pharmaceuticals, 2024, doi:10.3390/ph17081002_

Round 1

Reviewer 1 Report

Comments and Suggestions for Authors

See the attached paper

Author Response

Review for the article entitled

“ HPTLC Analysis of Anti-Asthmatic Combination Therapy in Pharmaceutical Formulation: Assessments of the Method's Greenness and Blueness”

(Manuscript ID: pharmaceuticals-2977608)

Authors:

Huda Salem AlSalem, Faisal K. Algethami, Maimana A. Magdy, Nourudin W. Ali, Hala E. Zaazaa,

  1. Abdelkawy, Maha M. Abdelrahman, Mohammed Gamal

submitted at Pharmaceuticals

After I read the article, I concluded that the article is well-organized, contain the components: Front matter (Title, Author list, Affiliations, Abstract, Keywords), Research manuscript sections (Introduction, Results and Discussion, Materials and Methods, Conclusions), Back matter (Author Contributions, Funding, Data Availability Statement, Acknowledgments, Declaration of generative AI and AI-assisted technologies in the writing process, Conflicts of Interest, References).

Below are the main points that I think should be reviewed and my comments.

Thanks for positive point of view. We appreciate your valuable comments and efforts

Abstract

Line 275: “…chloroform: ammonium acetate buffer...” Comment 1: Write “…chloroform : ammonium…”

Done

Introduction

Lines 297-298: “Hydroxyzine hydrochloride (HYX) chemically is identified as 297 (RS)-2-[2-[4- [(4-chlorophenyl)…”

Comment 2: I suggest: “The chemical name of hydroxyzine hydrochloride (HYX) is (RS)-2-[2-[4- [(4-chlorophenyl)…”

Done

Lines 301-302: “EPH, also known as Ephedrine Hydrochloride, is a chemical 301 compound with the chemical name (1R,2S)-2-(methylamino)…”

Comment 3: I suggest: “The chemical name of Ephedrine hydrochloride (EPH) is (1R,2S)-2- (methylamino)…”

Done

Lines 305-306: “Theophylline (THP), chemically identified as 1,3-dimethyl-3,7-dihydro…” Comment 4: I suggest: “The chemical name of Theophylline (THP) is 1,3-dimethyl-3,7-dihydro…”

Done

 Line 338: “about stationary materials, mobile systems…”

Comment 5: I suggest: “about stationary phase materials, mobile phase systems…”

Done

Results and discussions

Line 348: “…TLC plates [46][47].”

Comment 6: Write: “…TLC plates [46, 47].”

Done

Lines 353-415: “2.1. Method development and optimization”

Comment 7: Since section 2. Results and Discussions presents data related to the development of the method, especially the study of some factors that influence the separation and detection method (composition of the mobile phase, scanning wavelength, slit size, saturation period) I consider it necessary that these to be described in section “3. Material and method”, before paragraph “3.3. Chromatographic separation”. In this section, only the results and discussions related to these aspects should remain.

As the academic editor requested in the first revision, additional detailed discussions about method optimization parameters were required. Therefore, the details for this discussion are appropriate in section 2 results and discussions. While summary of chromatographic procedures should be stated in section 3.3 chromatographic separation without repetition of data. We think further discussion of these aspects would be lengthy and repetitive

Line 372: “An acceptable chromatogram was achieved. The recorded” Comment 8: I suggest you specify the elution time as well?

As per your note, we modify the text to be clearer. Retardation factors are  proper values for TLC chromatograms.

An acceptable chromatogram was achieved in terms of resolution and peak shape. The recorded retardation factors Rf for EPH, THP, and HYX were 0.15, 0.40, and 0.65, correspondingly, as illustrated in Fig. 2.

Lines 416-423: “2.2. Method application”

Comment 9: At lines 421-423 are given the equations for peak area versus concentration and the correlation coefficient. Usually the coefficient of determination is presented instead r2. Please, make the correction in these lines as well as at Lines 425-426: “The r symbol refers to the correlation coefficient.”.

As per your valuable note, they were corrected.

Comment 10: The position in which these data are included (Lines 416-426) in the text is not the most appropriate. I suggest moving this part to the end of the Linearity and range section (see Comment 12).

As per your valuable note, they were moved.

2.3. Validation of the TLC-densitometric method

Lines 428-429: “The method was validated in line with ICH standards for validation of analytical procedures [48].”

Comment 11: This phrase should be moved to section 3. Materials and Methods Description, before LINEARITY. That section will briefly present the way in which the validation parameters are studied (linearity and range, detection and quantification limits, repeatability, intermediate precision and accuracy), in accordance with the ICH Q2 (R1) or ICH Q2 R(2) standards (See also Comment 50).

We think that it should be noted the validation approach ICH or USP before stating extra details about validation items. Practical details about method validation were moved to section 3. Materials and Methods Description as recommended.

Linearity and Range

Line 431: “The calibration graphs linking the peak area and the EQUIVALENT concentrations of…”

Comment 12: I suggest: “The calibration graphs linking the peak area and the CORRESPONDING concentrations of…”

Done

Lines 436-438: Table 1

Comment 13: In Table 1, as it is now presented in the text, only the data related to linearity and range should remain, i.e. from …Range (mg/band)… to …The coefficient of determination. If you want to keep the table in its current form, I suggest you move it to the end of the validation section, before Robustness. But, in this position, insert Lines 417-426 (see Comment 10).

Lines 417 -426 were moved as suggested. The data in table 1 is usually represented as one table.  Data of linearity is mentioned first. So, position of table 1 is correct. 

Line 442: “Considered from the formulas [LOD =3.3 (SD / slope), LOQ = three times of LOD]”

Comment 14: Because the method validation has been assessed in line with international conference for harmonization (ICH) standards, I suggest: LOQ = 10 (SD/SLOPE), even if LOQ = three times of LOD = 3 x 3.3 (SD / slope) = 9.9 (SD/SLOPE)

As per your valuable note, it was corrected.

Line 443

Comment 15: I suggest move here the section Quantitation and detection limits (LOQ and LOD) calculations and present the calculation equations with the values for the Slope and Standard Deviation (SD) of the regression line. See also Comment 23.

Done

For HYX  as an example

LOD = 3.3 (SD of the response/slope) = 

= 3.3 × 0.0988 /3.6226 = 0.09

LOQ = 10 (SD of the response/slope)

= 10 × 0.0988 /3.6226= 0.27

Comment 16: Insert a paragraph with the description of the LOD and LOQ determination in section 3, after Linearity (see Comments 11, 23, 50).

Done

Accuracy

Lines 445-446: “The novel TLC method’s accuracy was further confirmed via the use of the standard addition approach.”

Comment 17: Because (i) the method was validated in line with ICH standards for validation of analytical procedures and (ii) at Lines 445-446 the authors show that the novel TLC method's accuracy was further confirmed via the use of the standard addition approach, should leave data in this sense. In ICH Q2 (R1) is stated that “The analytical procedure is applied to a matrix of all components except the analyte where a known amount of the analyte of interest has been added. In cases where all the expected components are impossible to reproduce, the analyte can be added to or enriched in the test sample. The results from measurements on unspiked and spiked/enriched samples are evaluated.

We apologize for this confusion. The accuracy of the novel HPTLC was assesses in three various approaches. The text was modified in section 2 and 3.  New 2 tables S1 and S2 were added in supplementary file.

  1. Accuracy

Detailed results of accuracy for the assay of pure powder samples of HYX, EPH, and THP using the novel HPTLC-densitometric approach are demonstrated in Table S1. Furthermore, the convenient recoveries of the standard addition approach assured the accuracy of the HPTLC approach, as described in Table S2. The novel HPTLC method was efficaciously applied for the analysis of HYX, EPH, and THP in their pharmaceutical tablets, and by applying the standard addition protocol, no interference from additives that may be found in the tablets was observed, as exposed in Table S2.

Morover, a statistical comparison of the outcomes for the novel TLC method and the reported HPLC method (RP-HPLC method using a C18 stationary phase and a mobile phase consisting of equal volumes of acetonitrile and 0.1% (w/v) aqueous ammonium carbonate buffer with pH 7 controlled with acetic acid at a rate of flow 2 mL/min) [40] demonstrates that the computed F- and t-values are less than the hypothetical ones, indicating that there are no noticeable distinctions between the two chromatographic techniques in terms of accuracy, as illustrated in Table 2.

Lines 449-452: Table 2

Comment 18: The data presented in table 2 are in accordance with Paragraph 3.3.1.3 Orthogonal procedure comparison of ICH Q2 R(21): “The results of the proposed analytical procedure are compared with those of an orthogonal procedure. The accuracy of the orthogonal procedure should be reported.”. If you have chosen this way of studying accuracy, I suggest that the previous paragraph (Lines 445-446) be corrected.

Done

Comment 19: Insert a paragraph with the description of the accuracy determination in section 3, before Application to Bronchaline® Tablets: Dosage Form (see Comments 11, 50)

Done

Lines 453-455: “The RP-HPLC method using a C18 column and liquid system consisted of equal amounts of acetonitrile and 0.1% (w/v) aqueous ammonium carbonate buffer solution adjusted to pH 7 with acetic acid at a movement speed of 2 mL/min.”

Comment 20: I suggest “The RP-HPLC method using a C18 column and a MOBILE PHASE CONSISTING of equal amounts of acetonitrile and 0.1% (w/v) aqueous ammonium carbonate buffer adjusted to pH 7 with acetic acid at a FLOW RATE of 2 mL/min.”

Done

Precision

Line 464: “duplicate assessments of the exact solutions”

Comment 21: What does “the exact solutions” mean? I think the word “exact” should be removed.

Done

Comment 22: Insert a paragraph with the description of the Precision (Repeatability and Intermediate precision) determination in section 3, before Application to Bronchaline® Tablets: Dosage Form (see Comments 11, 50)

Done

Quantitation and detection limits (LOQ and LOD) calculations

Lines 472-473: “LOQ and LOD were computed according to the next formulas. LOQ = (SD of the response/slope) × 10, while LOD = (SD of the response/slope) × 3.3.”

Comment 23: Insert a paragraph with the description of the LOD and LOQ determination in section 3, after Linearity (see Comments 11, 16, 50). I suggest you move this paragraph between the Linearity and Accuracy sections.

Done

Robustness

Line 475: “liquid chromatography [49][50].”

Comment 24: Write “liquid chromatography [49, 50].”

Done

Lines 492-493: “…C18 stationary phase and liquid system consisting of equal volumes of acetonitrile

…”

Comment 25: Write “…C18 stationary phase and A MOBILE PHASE consisting of equal volumes of acetonitrile …”

Done

Line 494: “… at a movement speed of 2 mL/min)…”

Comment 26: Write “… WITH A FLOW RATE of 2 mL/min)…”

Done

 Lines 491-497

Comment 27: The paragraph written on these lines has nothing to do with the robustness of the proposed TLC method. I suggest moving this paragraph to section “2. Results and Discussion”, “Accuracy”.

Done

Lines 499-511, Table 3

Comment 28: I suggest moving Table 3 before Line 484: “System integration evaluations are an essential component”.

Done

Lines 512-520

Comment 29: I suggest moving Table 4 and the paragraph from Lines 517-520 after Line 490: “and the selectivity is greater than one.”

Done

The greenness assessment for the novel TLC method

Lines 539: “contains 4 red subsections that are sections 1, 7, 13, 1, and 15.”

Comment 30: The number 1 appears twice. Is it a typing error or is it another number?

It was a typing error, corrected.

Lines 545-546, Lines 549-550, Lines 566-567: “with a mobile system of chloroform and ammonium acetate solution (9.5:0.5, v/v) adjusting to pH 6.5 using diluted ammonia solution.”

Comment 31: I suggest “with a mobile phase consisting of a mixture of chloroform and ammonium acetate solution (9.5:0.5, v/v) adjusted to pH = 6.5 using dilute ammonia solution.”

Done

The method's comparisons with the old reported HPLC methods and TLC ones

Lines 637-640, Table 5, In Rows “Run time/per sample / ≃ 1.5 min / ≃ 12 min / ≃ 15 min” and Comments / Short run time per sample / … / Moderately long run time “

Comment 32: What exactly does the time of ≃ 1.5 min refer to? Elution time? Scan time? Both?

Please specify.

This value was calculated as follows

Each TLC plate contain 20 samples [ three drugs in each sample ] , the elution time is 30 minutes. The run time per sample = 30/20 = 1.5 min. however, as per reviewer recommendation to include saturation time [ 20 minutes] it will be 50/20= 2.5.  The value was modified in table and text with clarifications in table 5.

Comment 33: How do you justify the statement that for the proposed TLC method a Short run time per sample (≃ 1.5 min) is obtained compared to the Reported HPLC method II [41] where a Moderately long run time (≃ 15 min) is obtained considering the fact that for the proposed method, is it necessary to saturate the chromatographic tank with the mobile phase of 20 min? Without this saturation time you would not be able to do the TLC separation... This means that this time is also part of the running time.

Saturation time was included as recommended while in HPLC methods in table 5, a sentence about normalization time was included [Without including the actual time required for normalizing the baseline ]

Table 6, Row 6: “Liquid system” Comment 34: I suggest “Mobile phase”.

Done

Comment 35: For consistency, use the same type of notation (eg “V/V” or “v/v”, “Silica gel” or “silica gel”, “60F254” or “60F254”, "Silica gel 60F254” or “60F254 silica gel”.

Done

Comment 36: In Table 6, row – Merits: Delete the line before “-Assay of real” in column 1, 4 and 9.

Done

 Comment 37: In Table 6, row – Demerits, column 8: Delete the underline after “Long degassing time

45 minutes _”

Done

Materials and methods description

Line 256: “3. Materials and Methods Description” Comment 38: I suggest only: “3. Materials and Methods”

Done

Line 258: “A short UV radiation wavelength of 254 nm (United States) was used for drug visibility” Comment 39: I don’t understand what the authors want to say with “(United States)”

Illustrated as recommended. Manufactured in United States.

Lines 260-261: “The spraying frequency was 10 s/μL.”

Comment 40: Frecquency in s/μL ? Maybe “The spraying frequency was 10 μL/s.”

Corrected

Line 271: “Their purities were 100.52, 99.85, and 99.32 for HYX, EPH and THP…”

Comment 41: Purities without measurement units? Maybe “Their purities were 100.52, 99.85, and 99.32% for HYX, EPH and THP…”

Corrected

Line 286: “2.3. Mother liquid preparations”

Comment 42: Change the number 2.3. with 3.3.

Corrected

Comment 43: It is not mandatory, but I suggest “Start solution…” instead of “Mother liquid…” Lines 287-288: “and THP (1 mg mL-1 in methyl alcohol);”

Comment 44: I suggest “and THP (1 mg/mL in methyl alcohol);” or “and THP (1 mg ´ mL-1 in methyl alcohol);”

Done

Line 290: “3.3 Chromatographic separation”

Comment 45: Change the number 3.3. with 3.4

Done

Lines 291-293: “Different aliquots of HYX, EPH, and THP were exactly relocated from their mother solutions into a set of 10-mL glass flasks. The capacity of each container was then filled with methyl alcohol.”

Comment 46: I understand, but something don’t sound good for me. I suggest “Different aliquots of HYX, EPH, and THP were exactly MEASURED from their START solutions and relocated into a set of 10 mL VOLUMETRIC flasks which were filled to the sign with methyl alcohol.”

Done

Lines 296-297: “in a chromatographic container that had been saturated for 20 minutes. The developing fluid system was made up of chloroform and ammonia acetate buffer solution in a volume ratio of 9.5:0.5…”

Comment 47: The chromatographic container had been saturated with what? I suppose with the mobile phase (developing fluid). In this case I suggest: “…in a chromatographic container that had been saturated for 20 minutes with the mobile phase (or, if you want, with the developing fluid system) WHICH IS A MIXTURE of chloroform and AMMONIUM acetate buffer solution in a volume ratio of 9.5:0.5…”. Take care: ammonium acetate, not ammonia acetate.

Done

Line 302: “3.4 Linearity”

Comment 48: Change the number 3.4. with 3.5.

Done

Lines 303-305: “Into two sets of 10 mL glass flasks, transfer perfectly the liquids equivalent to 0.4– 1.8, 2–16, and 0.4–1.8 mg from the aforementioned mother solution (1 mg mL-1) of HYX, EPH, and THP, then complete to the final size with methyl alcohol. Transfer 10 μL of each prepared solution to TLC sheets and develop under the stated settings in Section 2.1. Record the peak areas for HYX, EPH, and THP at 220 nm and create the calibration curves for the automatically calculated peak areas to the matching concentrations.”

Comment 49: The way this paragraph is written, it seems to be a work instruction, not a description of the method used. I suggest using past tense verbs. I suggest: “Volumes equivalent to 0.4– 1.8, 2–16, and 0.4–1.8 mg from the aforementioned start solution (1 mg/mL) of HYX, EPH, and THP were transferred into two sets of 10 mL volumetric flasks then completed to the final size with methyl alcohol. Volumes of 10 μL of each prepared solution were plotted to the TLC sheets and develop under the stated settings in Section 2.1. The peak areas for HYX, EPH, and THP are recorded at 220 nm and the calibration curves are created as the peak area versus the corresponding concentrations.”

Done

Comment 50: This section will briefly present the way in which the validation parameters are studied (linearity and range, detection and quantification limits, repeatability, intermediate precision and accuracy), in accordance with the ICH Q2 (R1) or ICH Q2 R(2). See also Comment 11.

Practical details about method validation were moved to section 3. Materials and Methods Description as recommended.

Line 309: “3.5 Application to Bronchaline® Tablets: Dosage Form” Comment 51: Change the number 3.5. with 3.6.

Done

Lines 310-311: “Ten Bronchaline® pills were well crushed and mixed. A carefully weighed amount of the tablets equal to 25 mg of HYX and 37.5 mg of EPH was moved to another 100-mL glass container; 75 mL of methyl alcohol was transferred and shaken for 20 minutes in a sonicator, filtered, and then the last size was used with methyl alcohol. This resultant solution is utilized as a stock standard solution for HYX (250 μg mL-1) and as a stock solution for EPH (375 μg mL-1), while another part of this liquid is diluted to get a stock solution for THP (1.5 mg mL-1) using methyl alcohol as a solvent.”

Comment 52: Why 10 tablets were well crushed and mixed? The authors should justify this number taking into account the mass of the tablets. Which is this mass? More, without this information, how we can weigh amount of the tablets equal to 25 mg of HYX and 37.5 mg of EPH? I suggest add to this section the required information, according to USP, British or European Pharmacopoeia, in order to know exactly how many tablets will be used. Please rephrase.

Done as suggested

The choice of 10 tablets in QC is a matter of sample size and statistical significance. By using a larger number of tablets, the sample becomes more representative of the overall batch, increasing the confidence in the test results. Using only one tablet may not provide a comprehensive understanding of the batch's characteristics. Probability of bias is higher. A larger sample size [10 tablets] reduces the impact of any potential variations within individual tablets. On the other hand, using a significantly higher number of tablets might not be practical due to wasting of resources and time. Some medications may have high prices and their destruction is a big lost. Therefore, Selecting 10 tablets for testing are a balance between obtaining a representative sample and ensuring efficient use of resources in the QC process. Besides, in all chapters of the British Pharmacopeia (BP) that discuss the analysis of tablet dosage forms, they select 10 tablets as a sample size such as the following cases

ASSAY

For tablets containing the equivalent of more than 2 mg of salbutamol

Carry out the method for liquid chromatography, Appendix III D, using the following solutions. Solution (1) contains 0.048% w/v of 2-tert-butylamino-1-(4-hydroxy-3-methylphenyl)ethanol sulphate BPCRS and 0.048% w/v of salbutamol sulphate BPCRS in methanol (10%). Solution (2) contains 0.048% w/v of salbutamol sulphate BPCRS in water. For solution (3) shake 10 tablets with 100 ml of water for 1 hour, add sufficient water to produce a solution containing the equivalent of 0.040% w/v of salbutamol, mix, centrifuge and use the supernatant liquid.

2- Content of cyanocobalamin, C63H88CoN14O14P

90.0 to 115.0% of the stated amount.

IDENTIFICATION

ıA. In the test for Uniformity of content, the principal peak in the chromatogram obtained with solution (1) shows a peak with the same retention time as the principal peak in the

chromatogram obtained with solution (2).ıB. Gently shake 10 tablets with 20 ml of chloroform to remove the coating and dry the tablet cores in a current of air. Transfer to a clean flask and powder with the aid of a glass rod. To a quantity of the powdered tablet cores containing 0.2 mg of cyanocobalamin, add 10 ml of a mixture of 1 volume of 2-ethoxyethanol and 3 volumes of water, shake vigorously for 5 minutes, centrifuge at 3000 revolutions per minute for 15 minutes and filter using a filter with a pore size of 0.45 μm. The light absorption of the filtrate, Appendix II B, in the range 345 to 560 nm exhibits maxima at about 361 nm and 550 nm. No maximum is exhibited at 351 nm (distinction from hydroxocobalamin).

Data are from British Pharmacopoeia 2013

As a common practice in British Pharmacopeia [1], ten tablets were selected to perform analysis of tablets dosage form.  Therefore, ten tablets which were used to avoid bias or variation between tablets and ensuring efficient use of resources in the QC processes.

The mass of one tablet was 0.26 g, and the mass of 10 tablets was 2.6 g. These 2.6 grams of the grinded tablets contained 100 mg of HYX, 150 mg of EPH, and 1200 mg of THP. A carefully weighed amount of about 0.65 g of tablet powder corresponding to 25 mg of HYX, 37.5 mg of EPH, and 300 mg of THP was dissolved in 75 mL of methyl alcohol and shaken for 20 minutes in a sonicator. The mixture was filtered, and the clear solution was diluted in a 100-mL volumetric flask with methyl alcohol. A part of this resultant solution is utilized as a start standard solution for HYX (250 μg/mL) and for EPH (375 μg/mL), and another part is diluted with methyl alcohol in a ratio of 1:1 to get a start solution for THP (1.5 mg/mL). 

Comment 53: This paragraph seems to be ambiguous:

  • The tablets do not contain only HYX and EPH. In paragraph b from section 2. Resources is stated for Bronchaline® pill to have 10 mg of HYX, 15 mg of EPH,

and 120 mg of THP. The quantities of HYX (25 mg) and EPH (37.5 mg) chosen by the authors are found in two and a half tablets. So, the quantity of tablets powder must be equal with 2.5 multiplied with the average mass of one tablet.

  • The powder are moved in a flask, diluted with methanol, shaken and filtered and after that “the last size was used with methyl alcohol”? It is not
  • … while another part of this liquid is diluted to get a stock solution for THP (1.5 mg mL-1)”. The dilution factor should be

New paragraph was provided including all reviewer recommendations. Thanks

Comment 54: In this context, the authors should explain how they choose the number of tablets used for obtaining the tablet powder and how they calculate the average mass of one tablet. At the end the paragraph should be something like: A carefully weighed amount about “a” g (and here they need the value = 2.5 x Average mass of one tablet) of tablets powder corresponding at 25 mg of HYX, 37.5 mg of EPH and 300 mg of THP was dissolved in 75 mL methyl alcohol and shaken for 20 minutes in a sonicator. The mixture was filtered and the clear solution was diluted in a 100 mL volumetric flask with methyl alcohol. A part of this resultant solution is utilized as a stock standard solution for HYX (250 μg/mL) and for EPH (375 μg/mL), and another part is diluted with methyl alcohol in a ratio of 1 / 1 to get a stock          solution for THP (1.5 mg/mL).

New paragraph was provided including all reviewer recommendations. Thanks

The mass of one tablet was 0.26 g, and the mass of 10 tablets was 2.6 g. These 2.6 grams of the grinded tablets contained 100 mg of HYX, 150 mg of EPH, and 1200 mg of THP. A carefully weighed amount of about 0.65 g of tablet powder corresponding to 25 mg of HYX, 37.5 mg of EPH, and 300 mg of THP was dissolved in 75 mL of methyl alcohol and shaken for 20 minutes in a sonicator. The mixture was filtered, and the clear solution was diluted in a 100-mL volumetric flask with methyl alcohol. A part of this resultant solution is utilized as a start standard solution for HYX (250 μg/mL) and for EPH (375 μg/mL), and another part is diluted with methyl alcohol in a ratio of 1:1 to get a start solution for THP (1.5 mg/mL). 

Conclusions

Lines 320-322: “For the first time, an outstanding TLC-densitometric approach in terms of sensitivity, functionality, and selectivity for the concurrent assay of HYX, EPH, and THP in their raw materials and pharmaceuticals.”

Comment 55: Something is missing in this sentence. Does not make sense. Please rephrase.

Done

 Lines 323-324: “an alternative to the old HPLC procedures mentioned above.”

Comment 56: I suggest “an alternative to the HPLC procedures mentioned above.”

Done

 Lines 319-330: Section – 4. Conclusions

Comment 57: Insufficient discussions on the results obtained. Is the functionality and applicability of the new TLC method proven only by a score of 77.5 through the Blue Applicability Grade Index (BAGI) approach? I suggest that in this section more aspects regarding the study results should be introduced, not just the general fact that it can replace HPLC methods.

Done

References

Comment 58: Please check the order of number in text. For example reference [63] pag 8 from manuscript appear after [40].

Reference numbers were corrected as recommended. Thanks

Comment 59: I believe that for a research article, out of a total of 63 bibliographic references, a number of only 23 bibliographic references (36.5%) published in the last 5 years is too small. I suggest the authors update the list of references.

The majority of the old references were related to old developed HPLC and TLC methods for the drugs separately or in combinations that we cannot ignore. However, 3 new references were provided in the last 5 years as recommended.

64- Mahmoud, S.A. and Abbas, A.E.F., 2024. Greenness, whiteness, and blueness assessment with spider chart solvents evaluation of HPTLC-densitometric method for quantifying a triple combination anti-Helicobacter pylori therapy. Sustainable Chemistry and Pharmacy, 37, p.101412.

65- Kamal, M.F., Abdel Moneim, M.M. and Hamdy, M.M., 2023. Green novel photometric and planar chromatographic assays of remdesivir: Comparative greenness assessment study using estimated GAPI tool versus ISO technical reported methods. Reviews in Analytical Chemistry, 42(1), p.20230060.

66- Parys, W., DoÅ‚owy, M. and Pyka-PajÄ…k, A., 2022. Significance of chromatographic techniques in pharmaceutical analysis. Processes, 10(1), p.172.

GENERAL CONCLUSIONS AFTER REVIEW

Not all the sections are well-developed.

As can be seen from my comments, some explanations are needed, the correction of some typing mistakes. However, I believe that there are many major aspects (clear presentation of work methods or protocols, presentation of calculation equations etc.) that must be resolved by the authors.

Section “Results and Discussions”

This section should contain the results obtained by the methods described in the “Materials and Methods Description” section, as well as their discussion/comment.

There are subsections related to the description / development / optimization of the method that do not correspond in the “Materials and Methods Description” section.

There are discrepancies between the data presented (for example, it is shown that, in order to perform TLC analysis, the saturation time (which is part of the running time) is 20 min and, on the other hand, a runtime / sample of ≃ 1.5 min is highlighted, much more smaller than in the case of other methods).

The results obtained during the validation of the method are not presented fluent, they are interspersed, some information that should be found in the “Materials and Methods Description” section is presented here.

There are discrepancies between the description of the method of determining the accuracy (by standard addition approach) and the experimental results (which seem to be performed through an orthogonal procedure).

All comments in section results and discussions were improved and clarified as recommended.

Section “Materials and Methods Description”

The method by which method development and optimization is carried out is not presented. The working method (TLC) is not sufficiently well described.

The way of working by which the validation of the method is carried out is not completely specified - although it is stated that it works according to ICH standards. There are some data related only to linearity, without presenting the method by which LOD, LOQ, repeatability, intermediate precision and accuracy are determined.

It is not specified the method by which the greenness assessment for the novel TLC method is carried out, as well as the method's practicality via the Blue Applicability Grade Index (BAGI) appraisal.

From this point of view, I suggest the revision of this section specifying the working conditions and parameters, a brief description of the method by which the validation is carried out as well as the method of determining the greenness assessment and the method's practicality via the BAGI appraisal.

All comments in section materials and methods were improved and clarified as recommended.

Section “Conclusions”

Insufficient discussions on the results obtained

More details were included in the conclusion as recommended.

Section “References”

Only 23 bibliographic references (36.5%) published in the last 5 years from a total of 63. In my opinion, it's a little bit.

The majority of the old references were related to old developed HPLC and TLC methods for the drugs separately or in combinations that we cannot ignore. However, 3 new references were provided in the last 5 years as recommended.

In conclusion, in order to meet the necessary conditions for publication, the manuscript requires a complete revision.

We replied to each point and highlighted changes in the text with green and turquoise colors

WE APPRECIATE THE REVIEWER FOR HIS PRECISE AND HELPFUL SUGGESTIONS.

Greetings.

Reviewer 2 Report

Comments and Suggestions for Authors

The paper “HPTLC Analysis of Anti-Asthmatic Combination Therapy in 257 Pharmaceutical Formulation: Assessments of the Method's 258 Greenness and Blueness” discusses the development and validation of a cost-effective, selective, and sensitive TLC-densitometric approach for the concurrent assay of Hydroxyzine Hydrochloride (HYX), Ephedrine Hydrochloride (EPH), and Theophylline (THP) in their pure form and pharmaceuticals. The innovative method utilizes a 60F254 silica gel stationary phase with a chloroform: ammonium acetate buffer as the mobile liquid system at 220 nm via UV detection. The method validation was conducted in accordance with international conference for harmonization (ICH) standards and was effectively used for the estimation of the researched medicines in their pharmaceuticals without intervention from excipients. The study also compared the new TLC method with previously published HPLC methods, highlighting its advantages in terms of short-run time per sample and moderate pH value for the liquid system.

Regarding comments and suggestions to improve the quality of publishing, the study could benefit from:

1.  A more detailed discussion on the limitations and challenges encountered during the method development and optimization. Additionally, providing a more comprehensive comparison with existing methods and discussing the potential implications of the findings for the pharmaceutical industry would enhance the overall quality of the study.

2.  Furthermore, the authors may consider expanding the discussion on the practical implications of the method, including its potential applications in pharmaceutical quality control and regulatory compliance. Additionally, providing a more in-depth analysis of the greenness assessment and its implications for sustainable analytical practices would further enrich the study.

3. The references provided are relevant and contribute to the contextualization of  the study. However, it would be beneficial to include more recent references to ensure the incorporation of the latest research in the field, for example:

a.  Caizhi Liao, Shadow Xiao, Xia Wang, Bench-to-bedside: Translational development landscape of biotechnology in healthcare,  Health Sciences Review, 7, 2023, 100097, https://doi.org/10.1016/j.hsr.2023.100097.

Also, it would be appreciated that if the authors could address the following questions:

1. What is the novel TLC-densitometric approach adapted for?

2. What parameters have been investigated that affect the chromatographic analysis?

3. What are the characteristics of the TLC approach regarding greenness?

4. How were the analytical parameters of the estimated HPTLC method compared to the reported HPLC methods?

Comments on the Quality of English Language

Moderate editing of English language required

Author Response

Reviewer 2:

The paper “HPTLC Analysis of Anti-Asthmatic Combination Therapy in 257 Pharmaceutical Formulation: Assessments of the Method's 258 Greenness and Blueness” discusses the development and validation of a cost-effective, selective, and sensitive TLC-densitometric approach for the concurrent assay of Hydroxyzine Hydrochloride (HYX), Ephedrine Hydrochloride (EPH), and Theophylline (THP) in their pure form and pharmaceuticals. The innovative method utilizes a 60F254 silica gel stationary phase with a chloroform: ammonium acetate buffer as the mobile liquid system at 220 nm via UV detection. The method validation was conducted in accordance with international conference for harmonization (ICH) standards and was effectively used for the estimation of the researched medicines in their pharmaceuticals without intervention from excipients. The study also compared the new TLC method with previously published HPLC methods, highlighting its advantages in terms of short-run time per sample and moderate pH value for the liquid system.

Thanks for positive point of view. We appreciate your valuable comments and efforts

Regarding comments and suggestions to improve the quality of publishing, the study could benefit from:

  1. A more detailed discussion on the limitations and challenges encountered during the method development and optimization. Additionally, providing a more comprehensive comparison with existing methods and discussing the potential implications of the findings for the pharmaceutical industry would enhance the overall quality of the study.

As per your valuable comment, a new section in page 9, 2.6 The Method's comparisons with the old reported HPLC methods and TLC ones was provided. Besides, the discussion and comparison with previous published methods were improved. New paragraph was provided at the end of the discussion illustrating the importance of the novel HPTLC method for QC and pharmaceutical industry.

By applying the novel HPTLC method specifically for HYX, EPH, and THP, analysts can achieve better separation, detection, and quantification of these drugs. This would lead to more accurate results, improving the quality of analysis in pharmaceutical research. The new HPTLC method can enhance the quality control process by enabling more precise measurements of pure drugs and pharmaceuticals. This would ensure that pharmaceutical products containing these substances meet the required specifications, reducing the risk of substandard or ineffective medications reaching asthmatic patients. Generally, this novel analytical approach would contribute to the pharmaceutical industry's ability to produce effective, safe, and high-quality pharmaceutical formulations.

66- Parys, W., DoÅ‚owy, M. and Pyka-PajÄ…k, A., 2022. Significance of chromatographic techniques in pharmaceutical analysis. Processes10(1), p.172.

67- Sonia, K. and Lakshmi, K.S., 2017. HPTLC method development and validation: An overview. Journal of Pharmaceutical Sciences and Research9(5), p.652.

  1. Furthermore, the authors may consider expanding the discussion on the practical implications of the method, including its potential applications in pharmaceutical quality control and regulatory compliance. Additionally, providing a more in-depth analysis of the greenness assessment and its implications for sustainable analytical practices would further enrich the study.

A new paragraph about greenness merits for the HPTLC method over the HPLC ones was provided illustrating the implication of sustainable analytical practices.

Notably, the greenness characteristics of the novel HPTLC approach compared to the HPLC one highlight several implications for sustainable analytical practices in the pharmaceutical industry. The lower solvent consumption, energy efficiency, minimized waste generation and its disposal, cost-effectiveness, instrument accessibility, and versatility associated with HPTLC contribute to sustainable analytical practices. Implementing these practices can lead to reduced environmental impact, lower costs, and improved operational efficiency in the pharmaceutical industry.

64- Mahmoud, S.A. and Abbas, A.E.F., 2024. Greenness, whiteness, and blueness assessment with spider chart solvents evaluation of HPTLC-densitometric method for quantifying a triple combination anti-Helicobacter pylori therapy. Sustainable Chemistry and Pharmacy37, p.101412.

65- Kamal, M.F., Abdel Moneim, M.M. and Hamdy, M.M., 2023. Green novel photometric and planar chromatographic assays of remdesivir: Comparative greenness assessment study using estimated GAPI tool versus ISO technical reported methods. Reviews in Analytical Chemistry42(1), p.20230060.

  1. The references provided are relevant and contribute to the contextualization of  the study. However, it would be beneficial to include more recent references to ensure the incorporation of the latest research in the field, for example:
  2. Caizhi Liao, Shadow Xiao, Xia Wang, Bench-to-bedside: Translational development landscape of biotechnology in healthcare,  Health Sciences Review, 7, 2023, 100097, https://doi.org/10.1016/j.hsr.2023.100097.

Thanks for your suggestion. The aforementioned article about Biotechnology which is may not fit our new HPTLC method. However, new updated 4 references were provided.

Also, it would be appreciated that if the authors could address the following questions:

  1. What is the novel TLC-densitometric approach adapted for?

This research work aims to provide a cost-effective, sensitive, selective, and reliable TLC-densitometric approach for the concurrent assay of HYX, EPH, and THP using the same chromatographic environment. Besides, TLC-densitometry can be utilized in quality control units as an inexpensive and time-efficient alternative to the costly HPLC method

  1. What parameters have been investigated that affect the chromatographic analysis?

Many parameters were discussed in details in section 2; such as Mobile phase, Scanning wavelength, Slit size, and Saturation period

  1. What are the characteristics of the TLC approach regarding greenness?

The greenness characteristics were discussed in details in section 2.4

As seen in Fig. 3, the TLC approach's AGREE [52] score of 0.56 reflects the low greenness attributes. Notably, the three red sub-sections 3, 10, and 11 refer to the off-line TLC situation, non-bio-based resources, and using hazard-mandatory chloroform.

Concerning GAPI outcomes, the resultant pictogram, Fig. 4, contains 4 red subsections that are sections 1, 7, 13, 1, and 15. These red sectors refer to offline assay, the use of chloroform hazard solvent, the possibility of chloroform vapors being released into the environment, and the non-treatment of solvent waste.

  1. How were the analytical parameters of the estimated HPTLC method compared to the reported HPLC methods?

Because no simultaneous HPTLC method was reported for the three drugs, we primarily compare our novel method with old HPLC ones. However, as per your value note and academic editor advice we add additional comparisons with old TLC methods.

Done as suggested. See the new section in pages 9and 10, 2.6 The Method's comparisons with the old reported HPLC methods and TLC ones.

Besides, new table 6 was provided entitled [Table 6: Comparisons between the novel HPTLC method and the old related ones for EPH, and THP in the review]

Thanks for your valuable comments and efforts
